# ADVERSARIAL-GUIDED DIFFUSION FOR ROBUST AND HIGH-FIDELITY MULTIMODAL LLM ATTACKS

## ABSTRACT

Recent diffusion-based adversarial attack methods have shown promising results in generating natural adversarial images. However, these methods often lack fidelity by inducing significant distortion on the original image with even small perturbations on the latent representation. In this paper, we propose Adversarial-Guided Diffusion (AGD), a novel diffusion-based generative adversarial attack framework, which introduces adversarial noise during the reverse sampling of conditional diffusion models. AGD uses editing-friendly inversion sampling to faithfully reconstruct images without significantly distorting them through gradients on the latent representation. In addition, AGD enhances latent representations by intelligently choosing sampling steps, thereby injecting adversarial semantics more smoothly. Extensive experiments demonstrate that our method outperforms state-of-the-art methods in both the effectiveness of generating adversarial images for targeted attacks on multimodal large language models (MLLMs) and image quality, successfully misleading the MLLM's responses. We argue that the security concerns surrounding the adversarial robustness of MLLMs deserve increased attention from the research community.

## 1 INTRODUCTION

With the exponential increase in data, computational resources, and model parameters, recent advancements in large language models (LLMs), particularly multimodal large language models (MLLMs), have achieved superior performance across various tasks, such as text-to-image and image-to-text generation, which highlights their promising potential for a wide range of applications (Li et al., 2023; Bao et al., 2023; Zhu et al., 2023; Liu et al., 2023). Although significant research efforts have been made to improve the alignment of LLMs, recent studies indicate that the introduction of visual modalities in MLLMs increases their vulnerability to adversarial attacks. Specifically, these MLLMs with visual structures could be easily misled by adversarial examples, which are generated by introducing imperceptible perturbations to clean images (Zhao et al., 2023; Cui et al., 2024; Luo et al., 2024). As a result, it is essential to thoroughly investigate the adversarial robustness of these MLLMs before their deployment, and proactively address potential security vulnerabilities.

For the research of the MLLMs adversarial robustness, compared to the white-box access scenario (Shayegani et al., 2024; Gao et al., 2024; Cui et al., 2024), the scenario where adversaries have only black-box system access seek to deceive the model into returning the targeted responses represents the most realistic and high-risk scenario (Zhao et al., 2023; Dong et al., 2023; Bailey et al., 2023). Existing methods such as AtackVLM (Zhao et al., 2023), is the first study to comprehensively examine the adversarial robustness of MLLMs under both black-box and targeted settings. This work employs query-based attacks with transfer-based priors. However, adversarial perturbation-based attacks frequently produce low-quality and unnatural adversarial images, the images differ greatly from the actual data distribution of natural images, as shown in Figure 1 which limits the effectiveness of robustness evaluations.

Recent research integrates adversarial example generation into the reverse process of diffusion models (Chen et al., 2023a; Dai et al., 2023; Chen et al., 2023c; Xue et al., 2024) to produce high-quality and realistic adversarial samples. AdvDiff (Dai et al., 2023)introduces adversarial guidance during the reverse diffusion process; however, its sampling begins from a standard Gaussian distribution, which does not guarantee high-quality reconstructions (see Figure 1). AdvDiffuser

Figure 1: Adversarial images are crafted by different methods. The first column denotes a clean image, other columns denote baselines and our method. We zoom in part of the lower right for a better view.

attempts to add PGD (Madry et al., 2018) to the latent variables at each sampling step through iterations (Chen et al., 2023b). While this enhances adversarial effectiveness, it influences the image quality and increases computational cost. In addition, ACA (Chen et al., 2023c) perturbs the initial latent images at the beginning of the reverse diffusion process. According to diffusion model principles (Mao et al., 2023), this seriously distorts the generated image.

To address the above challenges, we propose AGD, an attack framework based on text-to-image conditional diffusion models (Rombach et al., 2022). AGD introduces adversarial noise during the reverse sampling process of conditional diffusion models and effectively generates adversarial images for targeted attacks on MLLMs. Specifically, we employ an open-source text-to-image generation model to generate the target text into a target image. A surrogate model with the same visual encoder architecture as the MLLM is then used to obtain adversarial gradients. These gradients are injected into the noise prediction during the reverse diffusion process, iteratively modifying the images through sampling steps to generate adversarial images that mislead the MLLM's response toward the target text. Moreover, we incorporate edit friendly inversion to ensure faithful reconstruction of the adversarial samples. Inspired by truncated diffusion (Meng et al., 2022), we also employ a sampling strategy selecting specific timesteps for adversarial guidance to optimize adversarial guidance.

Our contributions are summarized as follows:

- We propose AGD, a novel framework for targeted adversarial attacks on multimodal large language models. By introducing adversarial noise guidance into the reverse sampling process of diffusion generative models, AGD effectively generates adversarial images for targeted attacks to perform targeted adversarial attacks on multimodal large language models.

- Considering edit friendly inversion sampling strategy with truncated diffusion, AGD achieves the generation of high-fidelity adversarial images. AGD provides a novel perspective for performing targeted adversarial attacks on multimodal large language models.

- Extensive experiments conducted across several state-of-the-art multimodal large language models demonstrate that AGD outperforms previous state-of-the-art attack methods, including diffusion-based models. AGD achieves superior performance in targeted adversarial attacks with higher-quality adversarial images generation.

## 2 RELATED WORK

**Adversarial attack on multimodal large language models.** Adversarial attacks typically function by introducing imperceptible perturbations into clean images, result in misleading the targeted responses (Goodfellow et al., 2014; Kurakin et al., 2018; Dong et al., 2018). In the case of MLLMs, robustness of MLLMs is highly dependent on their most vulnerable input visual modality. Attackers can exploit the inherent weaknesses within the model's visual structure to craft adversarial examples. Adversarial attacks on MLLMs are generally categorized into two types: black-box (Zhao et al., 2023; Dong et al., 2023; Bailey et al., 2023) and white-box attacks (Shayegani et al., 2024; Gao et al., 2024; Cui et al., 2024). According to the attack objectives, these can further be divided into untargeted (Schlarmann & Hein, 2023; Cui et al., 2024) and targeted attacks (Zhao et al., 2023; Wang et al., 2023). Compared to the white-box access scenario, the scenario where adversaries have only black-box access seek to deceive the model into returning the targeted responses represents the most

realistic and high-risk scenario. For the black-box access open-source MLLMs, AttackVLM (Zhao et al., 2023)provides a comprehensive evaluation of the robustness of them, specifically targeting models that are susceptible to adversarial attacks. Their study highlights the challenges associated with conducting targeted adversarial attacks on MLLMs. Our work focuses on targeted adversarial attacks for tasks involving visual question answering (VQA) and image captioning in MLLMs.

**Diffusion-based unrestricted adversarial attack.** Due to the $\ell_p$-norm distance is inadequate to capture how human perceive perturbation accurately (Chen et al., 2023b; Shamsabadi et al., 2020; Yuan et al., 2022), a number of unrestricted attack methods have been proposed to improve pixel-based attack methods. In recent, diffusion models have been introduced into adversarial attack research due to it's (Ho et al., 2020; Song et al., 2021a;b) capable of generating natural and diverse outputs. Diffusion-based unrestricted methods such as AdvDiff (Dai et al., 2023) and AdvDiffVLM Guo et al. (2024) incorporate adversarial guidance during the reverse diffusion process by injecting adversarial gradients, enabling the generation of adversarial examples. Similarly, AdvDiffuser Chen et al. (2023b) applies Projected Gradient Descent (PGD) Madry et al. (2018) within the reverse diffusion process, adding adversarial perturbations to the latent images at each sampling step. However, methods like AdvDiffVLM and AdvDiffuser, which inject adversarial semantics at each timestep, tend to significantly degrade the quality of the generated images. In addition, AdvDiff uses of a standard Gaussian distribution as the starting point for sampling limits the high-fidelity reconstruction of adversarial examples. In contrast, ACA Chen et al. (2023c) adds adversarial semantics into the latent images at the starting point of the sampling process through multiple iterative sampling steps, which leads to substantial deviations in the generated image content due to ACA modifies the latent variables at the beginning of sampling process.

**Image editing using diffusion models** Image editing is one of the most fundamental tasks in computer vision. Diffusion generative models are now being applied to image-to-image editing tasks (Brack et al., 2024; Couairon et al., 2023; Wallace et al., 2023; Hertz et al., 2023) since their powerful generative capabilities in text-to-image generation. SDEdit (Meng et al., 2022) achieves image editing by introducing noise at an intermediate step in the diffusion process. However, the resulting images often deviate significantly from the input, requiring a trade-off between realism and editing performance. Some image editing techniques address this by using additional masks to restrict changes to specific regions of the image (Hertz et al., 2023; Couairon et al., 2023; Chen et al., 2024). Recently, semantic image editing methods (Mokady et al., 2023; Huberman-Spiegelglas et al., 2024; Brack et al., 2024) relied on inverting the deterministic DDIM sampling process, where DDIM inversion identifies an initial noise vector to reconstruct the input image when diffused along with the prompt. Nonetheless, small errors will still incur at each timestep, often accumulating the result in deviations from the input, and require expensive optimization to correct error (Mokady et al., 2023). To address this, edit-friendly inversion (Huberman-Spiegelglas et al., 2024) extracts these noise mappings for any given image, obtaining an inverted image without requiring additional optimization. The adversarial attack method proposed in this work uses edit friendly inversion with sampling strategy truncated diffusion, ensuring high-fidelity and efficient generation of adversarial examples for MLLM attacks.

## 3 PRELIMINARIES

### 3.1 PROBLEM DEFINITION

Given a victim MLLM, $f$, a typical visual question answer or image-to-text task is defined by

$$f_\omega(\boldsymbol{x}, \boldsymbol{c}_{\text{in}}) = \boldsymbol{c}_{\text{out}}, \tag{1}$$

where $\omega$ is the model parameter, $\boldsymbol{x}$ is the input image, $\boldsymbol{c}_{\text{in}}$ is the input text, and $\boldsymbol{c}_{\text{out}}$ is response text. In image-to-text task $\boldsymbol{c}_{\text{in}}$ is a placeholder and $\boldsymbol{c}_{\text{out}}$ is the caption; in visual question answer tasks, $\boldsymbol{c}_{\text{in}}$ is the input prompt and $\boldsymbol{c}_{\text{out}}$ is the answer. This paper focuses on targeted adversarial attacks against MLLMs, aiming to generate adversarial images that mislead the model into responding with specific target text, which is defined as follows:

$$f_\theta(\boldsymbol{x}_{\text{adv}}, \boldsymbol{c}_{\text{in}}) = \boldsymbol{c}_{\text{tar}}, \tag{2}$$

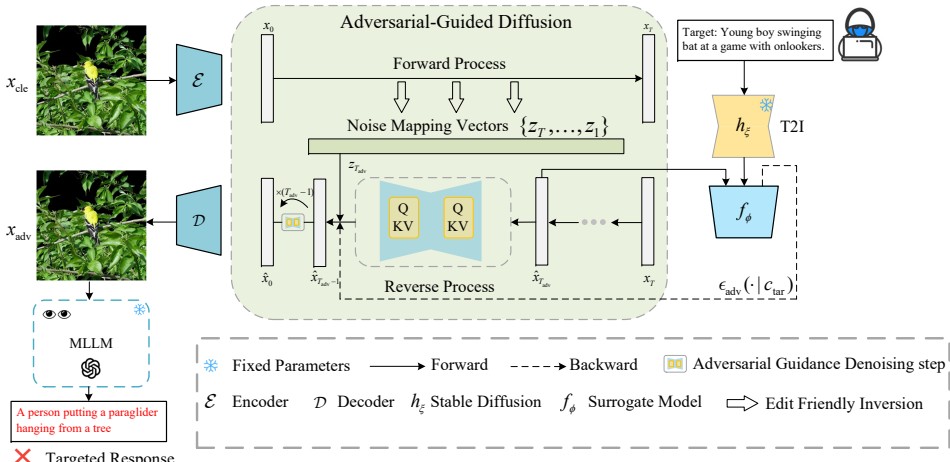

Figure 2: The main framework of our method. We adopt Stable Diffusion as our diffusion model. Firstly, we use edit friendly inversion to extract consistent noise maps in latent space. Next, adversarial-guided diffusion is used to generate adversarial examples by adding adversarial noise in the reverse sampling process. Finally, the generated adversarial examples are fed into victim MLLM resulting in targeted responses.

where $x_{\text{adv}}$ is the adversarial example, and $c_{\text{tar}}$ is the adversarial target text that the adversary expects the victim models to return. According to principles of adversarial attacks (Goodfellow et al., 2014), adversarial attacks on MLLMs summarized as the optimization of two objectives.

**Faithfulness.** The adversarial examples injected with adversarial semantics should be crafted in such a way that the responses generated by the victim multimodal large language model align with the target text.

**Fidelity.** The injection of adversarial semantics minimizes degradation of image quality, ensuring that the generated adversarial examples remain as visually similar to the original images as possible.

## 3.2 DIFFUSION PROBABILITY MODEL

As an effective generative model, Diffusion model (Ho et al., 2020) has be demonstrated that it generates images of higher quality and diversity than GANs (Dhariwal & Nichol, 2021). Diffusion models operate by defining a Markov chain and learning a denoising process to sample from a standard normal distribution $\mathcal{N}(x_T; \mathbf{0}, \mathbf{I})$. This process involves two phases: the forward process $q(x_t|x_{t-1}) := \mathcal{N}(x_t; \sqrt{1-\beta_t}x_{t-1}, \beta_t\mathbf{I})$ and the reverse $p_\theta(x_{t-1}|x_t) := \mathcal{N}(x_{t-1}; \mu_\theta(x_t, t), \Sigma_\theta(x_t, t))$. To improve upon this, Song et al. (2021a) introduced DDIM, which offers an alternative noise process not constrained by a Markov chain. By using the same training procedure as DDPM, DDIM enables faster sampling. In this paper, we use latent diffusion model (Rombach et al., 2022) with DDIM sampler as our diffusion generative model. As a effective conditional diffusion model, Stable Diffusion employs a classifier-free guidance that injects class information without relying on additional training of a classifier (Nichol et al., 2021; Ho & Salimans, 2021).

## 4 METHOD

### 4.1 ADVERSARIAL GUIDANCE NOISE PREDICTIONS

We display the whole framework of AGD in Figure 2, where we adopt the open-source Stable Diffusion (Rombach et al., 2022) as our diffusion model. Firstly, as discussion in §3.1, our objective of faithfulness is to leverage diffusion model to generate adversarial samples capable of successfully misleading MLLMs into targeted responses. Prior work demonstrates that classifier-guided diffusion sampling can serve as a gradient-based adversarial attack method (Dhariwal & Nichol, 2021).

Building upon this, we propose an adversarial-guided diffusion process that injects adversarial noise during the reverse sampling phase to generate adversarial examples with targeted semantic information. According to the definition of the score function in SDE (Song et al., 2021b) and Bayes' theorem,

$$
\begin{aligned}
\nabla_{\boldsymbol{x}_t} \log p_{\theta, f_\omega}(\boldsymbol{x}_t | \boldsymbol{c}_{\mathrm{tar}}) &= \nabla_{\boldsymbol{x}_t} \log \frac{p_\theta(\boldsymbol{c}_{\mathrm{tar}} | \boldsymbol{x}_t) p_{f_\omega}(\boldsymbol{x}_t)}{p_\theta(\boldsymbol{c}_{\mathrm{tar}})} \\
&= \nabla_{\boldsymbol{x}_t} \log p_\theta(\boldsymbol{x}_t) + \nabla_{\boldsymbol{x}} \log p_{f_\omega}(\boldsymbol{c}_{\mathrm{tar}} | \boldsymbol{x}_t),
\end{aligned}
\tag{3}
$$

where $\boldsymbol{x}_t$ represents the latent image of the diffusion model, the probability distribution $p_\theta(\boldsymbol{c}_{\mathrm{tar}})$ is independent of $\boldsymbol{x}_t$. Therefore, taking the gradient with respect to $\boldsymbol{x}_t$ results in zero.

For deterministic sampling methods like DDIM, we adopt score-based conditional diffusion, as proposed in (Song et al., 2021b). This approach leverages the inherent relationship between diffusion models and score matching. Specifically, if we have a model that can predict samples, denoted as $\epsilon_\theta(x)$, it can be utilized to derive the score function $\nabla_{\boldsymbol{x}_t} \log p_\theta(\boldsymbol{x}_t) = -\frac{1}{\sqrt{1-\bar{\alpha}_t}} \epsilon_\theta(\boldsymbol{x}_t)$. Substituting this into the score function as follows:

$$
\begin{aligned}
\nabla_{\boldsymbol{x}_t} \log(p_\theta(\boldsymbol{x}_t) p_{f_\omega}(\boldsymbol{c}_{\mathrm{tar}} | \boldsymbol{x}_t)) &= \nabla_{\boldsymbol{x}_t} \log p_\theta(\boldsymbol{x}_t) + \nabla_{\boldsymbol{x}_t} \log p_{f_\omega}(\boldsymbol{c}_{\mathrm{tar}} | \boldsymbol{x}_t) \\
&= -\frac{1}{\sqrt{1-\bar{\alpha}_t}} \epsilon_\theta(\boldsymbol{x}_t) + \nabla_{\boldsymbol{x}_t} \log p_{f_\omega}(\boldsymbol{c}_{\mathrm{tar}} | \boldsymbol{x}_t).
\end{aligned}
\tag{4}
$$

As introduction in §3.2, we can also obtain noise prediction based on text prompts. Finally, we define a new noise prediction take the form as follows,

$$
\tilde{\epsilon}_\theta(\boldsymbol{x}_t | \boldsymbol{c}, \boldsymbol{c}_{\mathrm{tar}}) = \hat{\epsilon}_\theta(\boldsymbol{x}_t | \boldsymbol{c}) - \sqrt{1-\bar{\alpha}_t} \nabla_{\boldsymbol{x}_t} \log p_{f_\omega}(\boldsymbol{c}_{\mathrm{tar}} | \boldsymbol{x}_t).
\tag{5}
$$

From this, we can deduce that adversarial condition diffusion generation can be interpreted as score guidance via an additional classifier gradient.

For adversarial guidance in the reverse diffusion process, we introduce adversarial perturbations beginning with $\hat{\boldsymbol{x}}_{T_{\mathrm{adv}}}$ rather beginning of sampling process noisy image $\boldsymbol{x}_T$, $T_{\mathrm{adv}}$ usually close to $\boldsymbol{x}_0$. This is because, during the reverse sampling process of diffusion models, the initial steps primarily focus on reconstructing the low-frequency contour information of the image. As shown in Figure 4, this reconstruction is crucial for maintaining the accurate global structure, which significantly influences the overall quality of the final generated image. To balance adversarial attack performance and image quality, we apply truncated diffusion techniques from image editing, selecting specific time steps for adversarial guidance (Meng et al., 2022; Huberman-Spiegelglas et al., 2024; Mao et al., 2023).

For the latent image $\boldsymbol{x}_t$ at time step $t$, the adversarial guidance noise prediction $\tilde{\epsilon}_\theta(\boldsymbol{x}_t | \boldsymbol{c}, \boldsymbol{c}_{\mathrm{tar}})$ is defined by Eq. 5. While the parameters of MLLMs $\omega$ are proprietary and inaccessible, attackers can reasonably be assumed to have knowledge of the visual encoders used in these models. This is because developers often disclose the architecture of visual encoders in technical reports, enabling the construction of surrogate models that utilize the same visual encoders for adversarial attacks. Then we maximize the following objective:

$$
\max f_\phi(\boldsymbol{x}_{\mathrm{adv}})^\top f_\phi(\boldsymbol{x}_{\mathrm{tar}}),
\tag{6}
$$

where $f_\phi$ is the surrogate model such as the CLIP (Radford et al., 2021) visual encoder, which is white-box accessibility and can obtain gradients through backpropagation. $\boldsymbol{x}_{\mathrm{tar}} = h_\xi(\boldsymbol{c}_{\mathrm{tar}})$ is the target image generated by target text $\boldsymbol{c}_{\mathrm{tar}}$ via a public text-to-image generative model such as Stabel Diffusion.

Thus, by maximizing objective Eq. 6, we approximate the adversarial score as follows:

$$
\nabla \log p_{f_\omega}(\boldsymbol{c}_{\mathrm{tar}} | \boldsymbol{x}_t) \simeq \nabla_{\boldsymbol{x}_t}(f_\phi(\boldsymbol{x}_{\mathrm{adv}})^\top f_\phi(\boldsymbol{x}_{\mathrm{tar}})),
\tag{7}
$$

Substituting this into Eq. 5, we obtain adversarial guidance noise prediction as follows:

$$
\tilde{\epsilon}_\theta(\boldsymbol{x}_t | \boldsymbol{c}, \boldsymbol{c}_{\mathrm{tar}}) = \hat{\epsilon}_\theta(\boldsymbol{x}_t | \boldsymbol{c}) + \boldsymbol{\epsilon}_{\mathrm{adv}}(\boldsymbol{x}_t | \boldsymbol{c}_{\mathrm{tar}}),
\tag{8}
$$

where $\boldsymbol{\epsilon}_{\mathrm{adv}}(\boldsymbol{x}_t | \boldsymbol{c}_{\mathrm{tar}}) = -\sqrt{1-\bar{\alpha}_t} \cdot s \cdot \mathrm{sign}(\nabla_{\boldsymbol{x}_t}(f_\phi(\boldsymbol{x}_{\mathrm{adv}})^\top f_\phi(\boldsymbol{x}_{\mathrm{tar}})))$, $s$ denotes scale parameter that controls the strength of adversarial guidance. According to Eq. 8, we apply the DDIM sampler to obtain the latent image $\boldsymbol{x}_{t-1}$ for the next time step.

We introduce momentum $m_t$ to accelerate guidance over timestep $t$ in the same target direction, the expression of the optimization adversarial surrogate model gradient $g_t$ as:

$$m_t \leftarrow \mu m_{t-1} + (1-\mu)g_t, \quad g_t = \frac{\nabla_{\boldsymbol{x}_t}(\hat{f}_\phi(\boldsymbol{x}_{\text{adv}})^\top f_\phi(\boldsymbol{x}_{\text{tar}}))}{\|\nabla_{\boldsymbol{x}_t}(\hat{f}_\phi(\boldsymbol{x}_{\text{adv}})^\top f_\phi(\boldsymbol{x}_{\text{tar}}))\|_1}, \quad (9)$$

where $\mu$ denote momentum factor and $\mu \in [0,1)$, with larger $\mu$ resulting in less volatile changes of the momentum. Moreover, the process is iterated $N$ times at each timestep $t$, yielding the final adversarial gradient that completes the current sampling step, as shown in Algorithm 1.

### 4.2 GENERATING NATURAL VISUAL ADVERSARIAL IMAGE VIA EDIT FRIENDLY INVERSION

As discussion in §3.1, the fidelity objective aims to minimize the discrepancy between the sampled reconstructed image and the original image. The generation of adversarial examples targeting MLLMs is similar to real image editing using diffusion models. Editing a real image using diffusion models requires extracting the noise vectors that would generate that image when used within the generative process (Mokady et al., 2023; Huberman-Spiegelglas et al., 2024). The edit friendly inversion (Huberman-Spiegelglas et al., 2024) method proposes a technique for extracting editing-friendly noise mappings for inversion, enabling precise image reconstruction.

For the DDIM sampling mentioned in §3.2, the generation process can be described as iteratively sampling from the random noise vector $\boldsymbol{x}_T \sim \mathcal{N}(\mathbf{0}, \mathbf{I})$ as follows:

$$\boldsymbol{x}_{t-1} = \hat{\mu}_t(\boldsymbol{x}_t) + \sigma_t \boldsymbol{z}_t, \quad t = T, \dots, 1. \quad (10)$$

The vector $\{\boldsymbol{x}_T, \boldsymbol{z}_T, \dots, \boldsymbol{z}_1\}$ uniquely determines the image $\boldsymbol{x}_0$ generated via Eq. 10. In other words, these vectors $\{\boldsymbol{x}_T, z_T, \dots, \boldsymbol{z}_1\}$ can be regarded as latent codes associated with the generated image. Edit friendly inversion aims to extract these noise vectors for a given real image $\boldsymbol{x}_0$, which are then used in Eq. 10 to reconstruct $\boldsymbol{x}_0$.

In fact, for any sequence of $T+1$ images $\boldsymbol{x}_0, \dots, \boldsymbol{x}_T$, where $\boldsymbol{x}_0$ represents a real image, consistent noise mappings can be extracted by isolating $\boldsymbol{z}_t$ from Eq. 10 as,

$$\boldsymbol{z}_t = \frac{\boldsymbol{x}_{t-1} - \hat{\mu}_t(\boldsymbol{x}_t)}{\sigma_t}, \quad t = T, \dots, 1, \quad (11)$$

we begin by determining the sequence $\boldsymbol{x}_0, \dots, \boldsymbol{x}_T$ based on $\boldsymbol{x}_0$, allowing us to extract the corresponding noise mapping vectors $\boldsymbol{x}_T, \boldsymbol{z}_T, \dots, \boldsymbol{z}_1$:

$$\boldsymbol{x}_t = \sqrt{\bar{\alpha}_t}\boldsymbol{x}_0 + \sqrt{1 - \bar{\alpha}_t}\tilde{\boldsymbol{\epsilon}}_t, \quad 1, \dots, T, \quad (12)$$

where $\tilde{\epsilon}_t \sim \mathcal{N}(\mathbf{0}, \mathbf{I})$ represents an independent noise, ensuring that $\boldsymbol{x}_t$ and $\boldsymbol{x}_{t-1}$ are further apart, resulting in each extracted $\boldsymbol{z}_t$ having a higher variance than in the standard generation process, which is more suitable for editing the global structure of the image. Finally, during the reverse sampling process, the extracted vector sequence $\{\boldsymbol{x}_T, \boldsymbol{z}_T, \dots, \boldsymbol{z}_1\}$, combined with Adversarial guidance noise predictions as described in §4.2, allows for high-quality image reconstruction through DDIM sampling in Eq. 10. We provide complete AGD algorithm in Algorithm 1.

## 5 EXPERIMENTS

### 5.1 EXPERIMENTAL SETTINGS

**Datasets.** The dataset consists of both images and prompts. Following (Zhao et al., 2023), We use validation set of ImageNet-1K as clean images, and we randomly select 1000 text descriptions from from MS-COCO captions (Lin et al., 2014) as our adversarial target texts.

**Victim MLLMs** In this paper, to evaluate the performance of our AGD on the MLLMs attack, We conducted targeted adversarial attack experiments on several advanced open-source multimodal large language models, including UniDiffuser (Bao et al., 2023), which employs a diffusion-based framework to jointly model the distribution of image-text pairs, enabling both image-to-text and text-to-image generation. BLIP-2 (Li et al., 2023), integrates a querying transformer and a large language model to boost image-grounded text generation. Furthermore, Img2Prompt (Guo et al., 2023) is designed to support zero-shot VQA tasks with a plug-and-play, LM-agnostic module. In recent, LLaVA (Liu et al., 2023) have scaled up the capabilities of large language models, utilizing Vicuna-13B (Chiang et al., 2023) to improve performance on image-grounded text generation tasks.

| MLLM | Method | Text encoder (pretrained) for evaluation | | | | | |
|------|--------|------|------|------|------|------|------|
| | | **RN50** | **RN101** | **ViT-B/16** | **ViT-B/32** | **ViT-L/14** | **Ensemble** |
| UniDiffuser | MF-it | 0.655 | 0.639 | 0.670 | 0.698 | 0.611 | 0.656 |
| | MF-ii | 0.709 | 0.695 | 0.722 | 0.733 | 0.637 | 0.700 |
| | AdvDiffuser | 0.427 | 0.429 | 0.453 | 0.472 | 0.338 | 0.424 |
| | ACA | 0.448 | 0.439 | 0.456 | 0.466 | 0.322 | 0.426 |
| | AGD(our) | **0.718** | **0.706** | **0.732** | **0.744** | **0.650** | **0.710** |
| Img2Prompt | MF-it | 0.499 | 0.472 | 0.501 | 0.525 | 0.355 | 0.470 |
| | MF-ii | 0.502 | 0.479 | 0.505 | 0.529 | 0.366 | 0.476 |
| | AdvDiffuser | 0.492 | 0.464 | 0.493 | 0.521 | 0.357 | 0.465 |
| | ACA | 0.502 | 0.479 | 0.505 | 0.525 | 0.358 | 0.473 |
| | AGD(our) | **0.505** | **0.481** | **0.509** | **0.531** | **0.367** | **0.479** |
| BLIP-2 | MF-it | 0.492 | 0.474 | 0.520 | 0.546 | 0.384 | 0.483 |
| | MF-ii | 0.562 | 0.541 | 0.573 | 0.592 | 0.449 | 0.543 |
| | AdvDiffuser | 0.457 | 0.469 | 0.468 | 0.457 | 0.356 | 0.448 |
| | ACA | 0.472 | 0.458 | 0.478 | 0.458 | 0.349 | 0.450 |
| | AGD(our) | **0.630** | **0.612** | **0.641** | **0.652** | **0.531** | **0.613** |
| LLaVA | MF-it | 0.389 | 0.441 | 0.417 | 0.452 | 0.288 | 0.397 |
| | MF-ii | 0.396 | 0.440 | 0.421 | 0.450 | 0.292 | 0.400 |
| | AdvDiffuser | 0.512 | 0.536 | 0.539 | 0.566 | 0.379 | 0.510 |
| | ACA | 0.538 | 0.507 | 0.542 | 0.565 | 0.386 | 0.507 |
| | AGD(our) | **0.542** | **0.510** | **0.547** | **0.572** | **0.393** | **0.513** |

Table 1: Comparison with state-of-the-art adversarial attack methods for performance of targeted attacks against victim MLLMs. We report the CLIP score $\uparrow$ between the generated responses of input images $x_{\text{adv}}$ and targeted texts $c_{\text{tar}}$, as computed by different CLIP text encoders and their ensemble/average results. The best result is bolded.

**Baselines.** To evaluate the performance of our method, we will compare it with existing attack methods in state-of-the-art multimodal large models with gray-box setting, including MF-it and MF-ii (Zhao et al., 2023), and state-of-the-art adversarial attack method based on diffusion model AdvDiffuser (Chen et al., 2023b) and ACA (Chen et al., 2023c).

**Evaluation metrics.** Following (Zhao et al., 2023), we adopt CLIP score (Hessel et al., 2021), which compares the responses generated by the victim models and predefined target texts. These scores are computed using different CLIP. Moreover, to assess the quality of adversarial examples, we employ three evaluation metrics: SSIM (Wang et al., 2004), LPIPS (Zhang et al., 2018), and PSNR (Hore & Ziou, 2010).

**Experimental Details.** We use clean images to generate adversarial images with fixed resolution 512. We set scale parameter $s = 6$, the number of iteration $N = 50$, momentum factor $\mu = 0.9$, and $T_{\text{adv}} = 5$. In addition, we use Stable Diffusion 2.1 (Rombach et al., 2022) with DDIM sampler (Song et al., 2021a)(the number of forward diffusion steps $T = 100$) to generate target images from the target texts, clean prompts are automatically generated using BLIP-2 (Li et al., 2023). In the experiments, we report the average CLIP score of 1000 adversarial images after evaluation on MLLMs and the average image evaluation metrics.

## 5.2 TARGETED ATTACK RESULTS ON MLLMS

As shown in Tabel 1, we evaluate effectiveness of our method on different victim MLLMs. Compared with recent targeted adversarial attack methods: MF-ii, MF-it, diffusion-based method AdvDiffuser, and ACA, experiment results demonstrate that our method consistently outperforms baselines in terms of CLIP score. Specifically, our method exhibit significant improvements of targeted attack such as UniDiffuser and BLIP-2. This observation indicates the effectiveness of our methods targeted attack against victim MLLMs.

| Method | UniDiffuser | | | BLIP-2 | | | LLaVA | | |
|---|---|---|---|---|---|---|---|---|---|
| | SSIM ↑ | LPIPS ↓ | PSNR ↑ | SSIM ↑ | LPIPS ↓ | PSNR ↑ | SSIM ↑ | LPIPS ↓ | PSNR ↑ |
| MF-it | 0.4322 | 0.5028 | 17.01 | 0.4428 | 0.4930 | 17.06 | 0.4239 | 0.4912 | 17.05 |
| MF-ii | 0.4342 | 0.4987 | 17.01 | 0.4342 | 0.4987 | 17.05 | 0.4368 | 0.4943 | 17.05 |
| AdvDiffuser | 0.3706 | 0.7897 | 6.816 | 0.2572 | 0.7566 | 5.699 | 0.2562 | 0.7477 | 5.712 |
| ACA | 0.4310 | 0.5320 | 16.54 | 0.4335 | 0.5238 | 16.61 | 0.4384 | 0.5172 | 16.75 |
| AGD(our) | **0.4579** | **0.3293** | **17.89** | **0.4512** | **0.3193** | 16.96 | **0.4539** | **0.3291** | **17.50** |

Table 2: Performance comparison of different methods based on SSIM, LPIPS, and PSNR metrics.

## 5.3 VISUALIZATION

**Quantitative Comparison**    To evaluate the image quality of adversarial images generated by our method, we quantitatively assess the image quality using image quality evaluation metrics such as SSIM, LPIPS, and PNSR. As illustrated in Table 2, compared to baselines, the adversarial images generated by our method exhibit higher image quality, especially on LPIPS. This is attributable to the fact that we apply adversarial noise guidance while strategically selecting the timesteps for adversarial semantics injection during the sampling process, based on the concept of truncated diffusion.

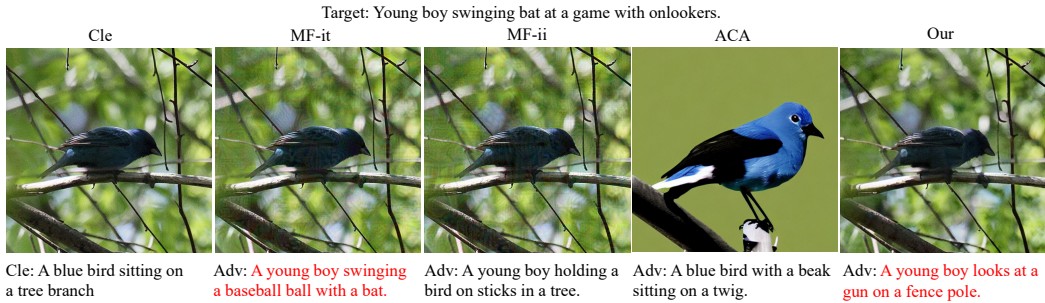

Figure 3: Comparison of different targeted adversarial attacks and our method on UniDiffuser. We provide clean image, images generated by MF-ii, MF-it (Zhao et al., 2023), ACA (Chen et al., 2023c), and our method. In addition to the visualization of adversarial examples, we display the adversarial target text above the image and show the caption results for both the original image and the adversarial example from different baselines below the image.

**Qualitative Comparison**    We visualize adversarial images generated by our method and other bashlines. As shown in Figure 3, compared to the adversarial images generated by baselines, our method substantially preserves the structure and natural appearance of the clean images. In contrast, MF-it and MF-ii directly introduce adversarial perturbations in terms of $\ell_p$-norm limitation to the clean images. Furthermore, ACA significantly changes the image structure by introducing adversarial perturbations to latent during the early stages of the reverse process. Moreover, we present the responses from MLLMs when input adversarial images are generated by different methods, demonstrating that our method successfully misleads the MLLM's response (More results see Appendix D.1).

## 5.4 ABLATION STUDY

**The impacts of hyperparameters.**    We first explore the impact of hyperparameter adversarial scale $s$, inner iterations $N$, and momentum factor $\mu$. We conduct experiments on UniDiffuser with $s$ in a range of [0.5, 7.0] with 0.5 intervals, other hyperparameters are the same as the above targeted attack experiments. As shown in Figure 5a, we report the average CLIP score vs. LPIPS similarity trade-off. The results show that increasing $s$ enhances attack performance but diminishes the visual quality of adversarial examples, our method improves the attack performance and influences the image quality in a small range. Similarly, We conduct experiments on UniDiffuser with $N$ varies in a range of [5, 55] with 5 intervals and $\mu$ in a range of [0, 0.9] with 0.1 intervals. From the results in Figure 5c, we find that larger values for $N$ result in a greater CLIP score, but it does not seriously

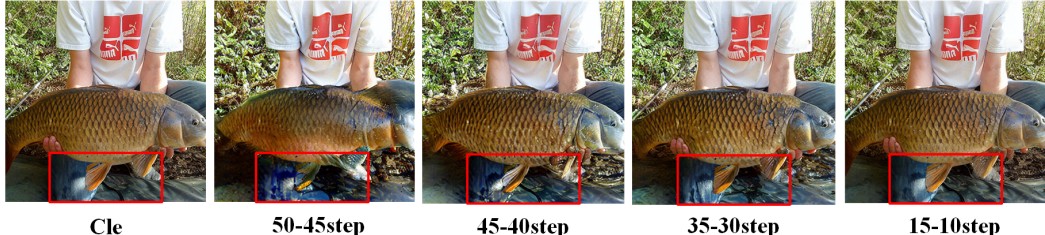

Figure 4: Visualization of generated adversarial images crafted by AGD when selecting different phases in the reverse sampling process. The first column displays the original images, while the other columns illustrate the effects of adversarial guidance applied at different phases for fixed timesteps. We highlight the regions where significant structural changes are observed.

influence the quality of the generated adversarial images. This is because, in the inner loop, we update the adversarial surrogate model gradient to find the best adversarial guidance. From the results in Figure 5d, it's obvious that momentum is essential for adversarial guidance, especially bigger momentum factor $\mu$ results in greater attack performance.

**The impacts of sampling strategy** We explore the effects of the sampling strategy, as illustrated in Figure 5b. The results demonstrate that the CLIP score improves as $T_{adv}$ increases. This is attributed to the stronger adversarial guidance during the reverse sampling process. Furthermore, We explore the impact of attack phase selection on the quality of image generation. In the experiments, we choose different phases to inject adversarial guidance in the reverse sampling process in a total of 5 steps. From Figure 4, we find that the image's fidelity will be seriously influenced by the variance of image structure information if the introduction of adversarial guidance is early in the sampling process. This proves the rationality behind the sampling strategy employed in our AGD method for selecting timesteps when introducing adversarial guidance in our method.

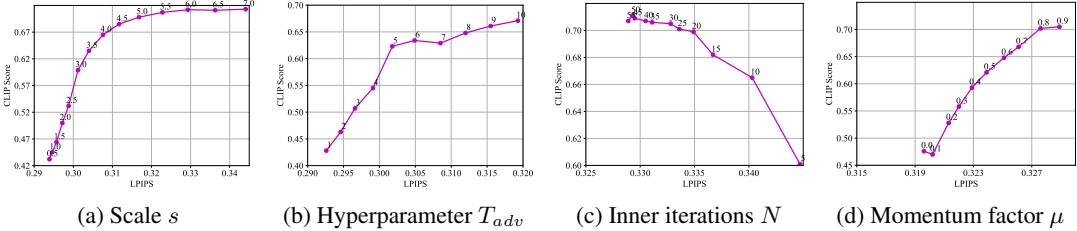

| (a) Scale $s$ | (b) Hyperparameter $T_{adv}$ | (c) Inner iterations $N$ | (d) Momentum factor $\mu$ |

Figure 5: Ablation study of the impact of hyperparameters. We plot CLIP scores (higher is better) of the target attributes against LPIPS similarity (lower is better).

## 6 CONCLUSION

In this paper, focusing on targeted adversarial attacks on MLLMs, we propose a diffusion-based adversarial attack framework AGD that addresses the key challenges of robust and high-fidelity multimodal LLM attacks. By introducing adversarial noise during the reverse sampling process and employing edit friendly inversion and selection of sampling strategy, our method improves image fidelity and adversarial effectiveness. Experimental results demonstrate superior performance over existing methods in generating high-fidelity adversarial images that successfully mislead MLLM responses, underscoring the need for further exploration of adversarial robustness in multimodal systems. Our work underscores the critical need to enhance robust evaluation techniques in order to mitigate security risks in MLLM applications, which will also guide future exploration in assessing MLLMs' vulnerability.

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

APPENDIX

## A   IMPLEMENTATION DETAILS

In our experiments, we experiment on the clean images to generate adversarial images with fixed resolution 512. We set scale parameter $s = 6$, the number of iteration $N = 50$, momentum factor $\mu = 0.1$, and $T_{\text{adv}} = 5$. In addition, we use Stable Diffusion 2.1 (Rombach et al., 2022) with DDIM sampler (Song et al., 2021a)(the number of forward diffusion steps $T = 100$) to generate target images from the target texts, clean prompts are automatically generated using BLIP-2 (Li et al., 2023). For prompt of querying LLaVA and Img2Prompt, we provide prompt fixed to be "what is the content of this image?". We also provide targeted images generated from targeted text in Figure 7.

For victim MLLMs in targeted attack experiment, we choose CLIP with ViT-B/32 vision encoder as surrogate model for UniDiffuser and Img2Prompt, CLIP with ViT-G/14 vision encoder as surrogate model for Img2Prompt BLIP-2, and CLIP with ViT-L/14 vision encoder as surrogate model for LLaVA.

## B   ALGORITHM

We provide a complete AGD algorithm in Algorithm 1. Firstly, we employ edit friendly inversion (Huberman-Spiegelglas et al., 2024) to obtain the noise mapping vector $x_T, z_T, \ldots, z_1$, ensuring that the clean image $x$ can be faithfully reconstructed. Building on this, we introduce our adversarial guidance noise predictions, integrated with a DDIM sampler to generate adversarial images with targeted semantic features. In Algorithm 1, we utilize momentum-based updates on the surrogate model to compute adversarial gradients. The adversarial guidance noise predictions, refined through multiple iterative updates at selected time steps, are then applied during the sampling process to optimize the adversarial image generation.

---

**Algorithm 1** The algorithm of AGD.

---

**Input:** Clean image $x_{\text{cle}}$, surrogate model $f_\phi$, target image $x_{\text{tar}}$, clean image caption $c$, momentum factor $\mu$, adversarial guidance scale $s$, atack start tiemstep $T_{adv}$, and number of iterations $N$

**Initialization:** momentum $m = 0$, inner momentum $\hat{m} = 0$, consistent noise maps vectors $\mathbf{V} = \{\}$, forward sequence $\mathbf{S} = \{\}$, and $x_0 = x_{\text{cle}}$

1: Add noise to $x_0$ obtain $\mathbf{S}$ via forward process Eq. 12
2: **for** $x_t \in \mathbf{S}$ **do**
3:     Predict $x_{t-1}$ by Eq. 10, then obtain noise map $z_t$ by Eq. 11
4:     $\mathbf{V} = \mathbf{V} \cup \{z_t\}$
5: **end for**
6: **for** $t = T, \ldots, T_{adv}, \ldots, 1$ **do**
7:     **if** $t \le T_{adv}$ **then**
8:         $\hat{m}_0 = m_t$
9:         **for** $i = 1, \ldots, N$ **do**
10:            Obtain gradient $g_i$ by $f_\phi$, $x_{\text{tar}}$, and $\hat{x}_{t-1}$
11:            $\hat{m}_i \leftarrow \mu \hat{m}_{i-1} + (1 - \mu) g_i$
12:            $\tilde{\epsilon}_\theta(\hat{x}_t | c, c_{\text{tar}}) = \hat{\epsilon}_\theta(\hat{x}_t | c) - \sqrt{1 - \bar{\alpha}_t} \cdot s \cdot \text{sign}(\hat{m}_i)$
13:            DDIM sampling $\hat{x}_{t-1}$ with $\tilde{\epsilon}_\theta(x_t | c, c_{\text{tar}})$ and noise map $z_t$
14:         **end for**
15:         $\tilde{\epsilon}_\theta(\hat{x}_t | c, c_{\text{tar}}) = \hat{\epsilon}_\theta(\hat{x}_t | c) - \sqrt{1 - \bar{\alpha}_t} \cdot s \cdot \text{sign}(\hat{m}_N)$
16:         DDIM sampling $\hat{x}_{t-1}$ with $\tilde{\epsilon}_\theta(x_t | c, c_{\text{tar}})$ and noise map $z_t$.
17:         $m_{t-1} \leftarrow \mu m_t + (1 - \mu) g_N$
18:     **else**
19:         DDIM sampling $\hat{x}_{t-1}$ with $\hat{\epsilon}_\theta(x_t | c)$ and noise map $z_t$.
20:     **end if**
21: **end for**
22: **Output:** $x_{\text{adv}} = \hat{x}_0$

---

# C ADDITIONAL DERIVATIONS

## C.1 DETAILED DERIVATION OF ADVERSARIAL GUIDANCE NOISE PREDICTIONS

For the diffusion process forward SDE (Song et al., 2021b):

$$d\boldsymbol{x} = \mathbf{f}(\boldsymbol{x}, t)dt + g(t)d\mathbf{w} . \tag{13}$$

For the reverse-time SDE:

$$d\boldsymbol{x} = [\mathbf{f}(\boldsymbol{x}, t) - g(t)^2 \nabla_{\boldsymbol{x}} \log p_t(\boldsymbol{x})]dt + g(t)d\overline{\mathbf{w}}, \tag{14}$$

Then we obtain conditional generation reverse-time SDE as follows:

$$d\boldsymbol{x} = \left[\mathbf{f}(\boldsymbol{x}, t) - g(t)^2 \nabla_x \log p_t(\boldsymbol{x}|\boldsymbol{c}_{\text{tar}})\right] dt + g(t)d\overline{\mathbf{w}} . \tag{15}$$

Using the Bayes' theorem,

$$p_{\theta, f_\omega}(\boldsymbol{x}_t|\boldsymbol{c}_{\text{tar}}) = \frac{p_\theta(\boldsymbol{c}_{\text{tar}}|\boldsymbol{x}_t) \, p_{f_\omega}(\boldsymbol{x}_t)}{p_\theta(\boldsymbol{c}_{\text{tar}})} , \tag{16}$$

then taking the gradient of the logarithm w.r.t. $\boldsymbol{x}_t$:

$$
\begin{aligned}
\nabla_{\boldsymbol{x}_t} \log p_{\theta, f_\omega}(\boldsymbol{x}_t|\boldsymbol{c}_{\text{tar}}) &= \nabla_{\boldsymbol{x}_t} \log \frac{p_\theta(\boldsymbol{c}_{\text{tar}}|\boldsymbol{x}_t) \, p_{f_\omega}(\boldsymbol{x}_t)}{p_\theta(\boldsymbol{c}_{\text{tar}})} \\
&= \nabla_{\boldsymbol{x}_t} \log p_\theta(\boldsymbol{x}_t) + \nabla_{\boldsymbol{x}_t} \log p_{f_\omega}(\boldsymbol{c}_{\text{tar}}|\boldsymbol{x}_t) - \nabla_{\boldsymbol{x}_t} \log p_\theta(\boldsymbol{c}_{\text{tar}}) \\
&= \nabla_{\boldsymbol{x}_t} \log p_\theta(\boldsymbol{x}_t) + \nabla_{\boldsymbol{x}_t} \log p_{f_\omega}(\boldsymbol{c}_{\text{tar}}|\boldsymbol{x}_t) .
\end{aligned}
\tag{17}
$$

# D ADDITIONAL EXPERIMENTS

## D.1 VISUALIZATION OF THE TARGETED ATTACK RESULTS

We visualize adversarial images generated by our method and other baselines on different victim MLLMs. As shown in Figure 6, compared to the adversarial images generated by baselines, our method substantially preserves the structure and natural appearance of the clean images. In contrast, MF-it and MF-ii directly introduce adversarial perturbations in terms of $\ell_p$-norm limitation to the clean images, ACA significantly changes the image structure by introducing adversarial perturbations to latent during the early stages of the reverse process. Furthermore, we present the responses from MLLMs when input adversarial images are generated by different methods. In a nutshell, our method consistently surpasses baselines across various MLLMs in terms of both generated image quality and attack results.

## D.2 LEVERAGING AGD FOR ENSEMBLE ATTACKS

Furthermore, we conduct a experment to compare ensemble attacks founded on our method with founded on other baselines. For ensemble attacks, we compute adversarial gradients $g_t$ as follows:

$$g_t = \frac{\nabla_{\boldsymbol{x}_t} \sum_{i=1}^{N_m} (\hat{f}_{\phi,i}(\boldsymbol{x}_{\text{adv}})^\top f_{\phi,i}(\boldsymbol{x}_{\text{tar}}))}{\|\nabla_{\boldsymbol{x}_t} \sum_{i=1}^{N_m} (\hat{f}_{\phi,i}(\boldsymbol{x}_{\text{adv}})^\top f_{\phi,i}(\boldsymbol{x}_{\text{tar}}))\|_1} , \tag{18}$$

where $N_m$ is the number of surrogate models. As shown in Table 3, our AGD achieved better results transferability when using conventional ensemble attacks strategy. These findings uncover the potential of our ADG for constructing ensemble adversarial attacks on MLLMs.

## D.3 COMPARISON OF CLIP-LPIPS TRADE-OFF FOR DIFFERENT ATTACK METHODS

The results in Figure 8 show that the CLIP-LPIPS trade-off of different attack methods on UniDiffuser. The top left corner represents the ideal attack method with maximum target semantics alignment without deviating from the initial image. It's obvious that our method is closest to the ideal region when CLIP score is high, demonstrating advantages of our method for robust and high-fidelity MLLM Attacks.

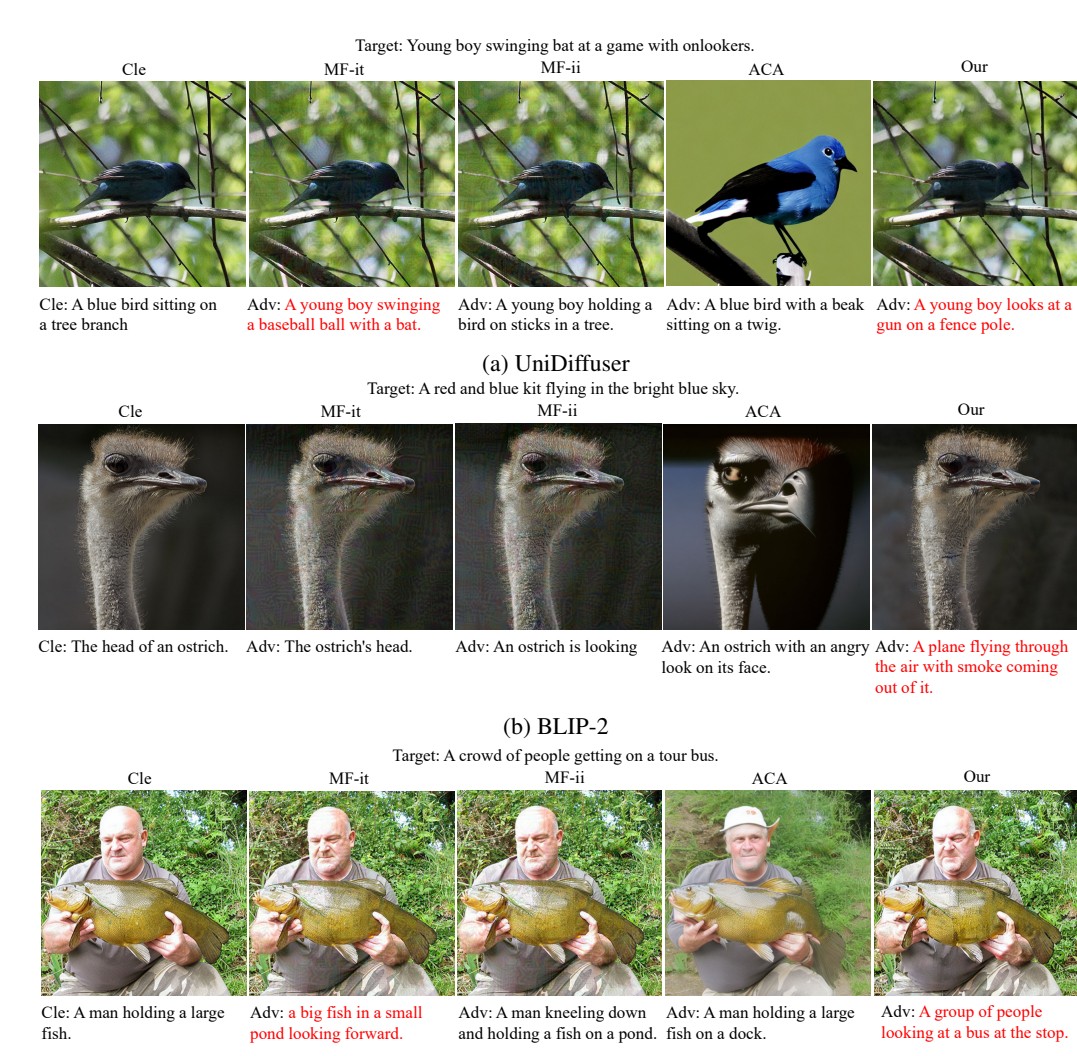

Figure 6: Comparison of different targeted adversarial attacks and our method on different MLLMs. e provide clean image, images generated by MF-ii, MF-it (Zhao et al., 2023), ACA (Chen et al., 2023c), and our method. In addition to the visualization of adversarial examples, we display the adversarial target text above the image and show the caption results for both the original image and the adversarial example from different baselines below the image.

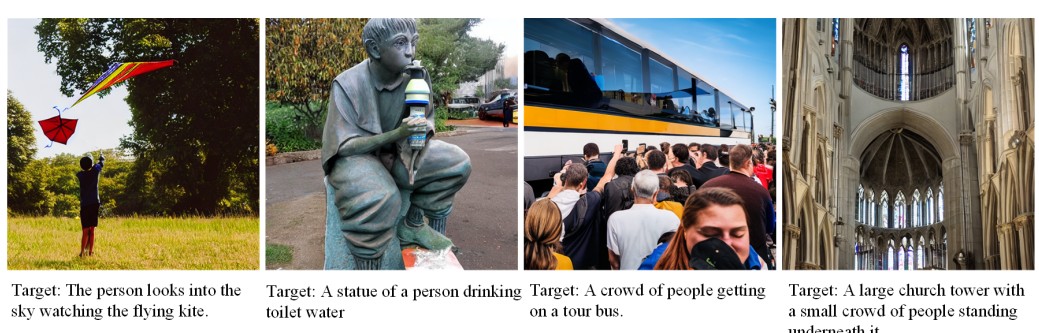

Figure 7: An illustration of target images generated from target text by Stable Diffusion.

| MLLM | Method | Text encoder (pretrained) for evaluation | | | | | | LPIPS↓ |
|---|---|---|---|---|---|---|---|---|
| | | RN50 | RN101 | ViT-B/16 | ViT-B/32 | ViT-L/14 | Ensemble | |
| UniDiffuser | MF-it | 0.638 | 0.626 | 0.652 | 0.668 | 0.550 | 0.627 | 0.3636 |
| | MF-ii | 0.689 | 0.675 | 0.702 | 0.715 | 0.614 | 0.679 | 0.3642 |
| | AGD(our) | **0.717** | **0.704** | **0.728** | **0.741** | **0.647** | **0.707** | **0.3313** |
| Img2Prompt | MF-it | 0.506 | 0.480 | 0.511 | 0.531 | 0.366 | 0.479 | 0.3636 |
| | MF-ii | 0.505 | 0.479 | 0.510 | 0.531 | 0.361 | 0.477 | 0.3642 |
| | AGD(our) | **0.508** | **0.483** | **0.514** | **0.533** | **0.368** | **0.481** | **0.3313** |
| BLIP-2 | MF-it | 0.476 | 0.459 | 0.487 | 0.507 | 0.358 | 0.458 | 0.3636 |
| | MF-ii | 0.486 | 0.464 | 0.494 | 0.514 | 0.364 | 0.464 | 0.3642 |
| | AGD(our) | **0.630** | **0.612** | **0.641** | **0.652** | **0.531** | **0.613** | **0.3313** |
| LLaVA | MF-it | 0.538 | 0.508 | 0.546 | 0.568 | 0.390 | 0.510 | 0.3636 |
| | MF-ii | 0.537 | 0.509 | 0.545 | 0.569 | 0.391 | 0.510 | 0.3642 |
| | AGD(our) | **0.543** | **0.514** | **0.550** | **0.573** | **0.397** | **0.515** | **0.3313** |

Table 3: Ensemble Attacks by our method. We report the CLIP score ↑ between the generated responses of input images $x_{\text{adv}}$ and targeted texts $c_{\text{tar}}$, as computed by different CLIP text encoders and their ensemble/average results. We also provide LPIPS to compare image quailty. The best result is bolded.

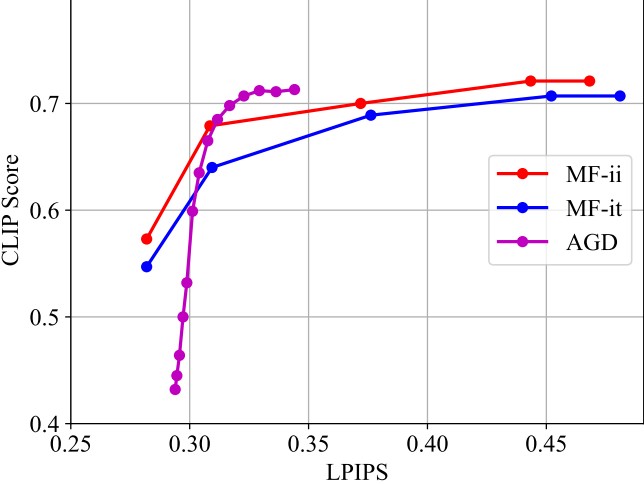

Figure 8: Comparison of the CLIP-LPIPS trade-off of different attack methods on UniDiffuser. We plot CLIP scores (higher is better) of the target text against LPIPS similarity (lower is better).

