# OpenReview forum: "Adversarial-Guided Diffusion for Robust and High-Fidelity Multimodal LLM Attacks"
_ICLR.cc/2025/Conference — ICLR 2025 Conference Withdrawn Submission_

### Official Review · Reviewer_p5rk · 2024-10-20

**Soundness:** 3
**Presentation:** 3
**Contribution:** 2
**Rating:** 5
**Confidence:** 4

**Summary:**

This paper proposes a novel method to generate high-fidelity adversarial examples for MLLMs. Specifically, it first defines an adversarial denoising step to inject adversarial signals into the image. Then, it exploits the a reverse sampling process similar to that in image editing methods to generate adversarial examples with high similarities to the original images.

**Strengths:**

1) The proposed method generates adversarial examples on MLLMs with high similarities to the base images. From the perspective of original-adversarial similarities, the proposed method outperforms existing methods without doubts.

2) It is an interesting idea to use the reverse sampling editing to make adversarial examples stick to the original images.

3) The generalizability of adversarial examples is impressive for they work in attacking different MLLMs.

**Weaknesses:**

1) Motivation: The introduction indicates that currently attacks fail in generating high-quality adversarial examples. However, according to the Figure 1, the exact situation is that these existing attacks fail in *reconstructing* the original image when generating adversarial examples. It is an interesting idea to do this reconstruction in adversarial attacks. However, it is not necessary because we only need the adversarial examples to be plausible rather than sticking to the content of some real images.  From my point of view, the ACA examples are good enough to escape from human vision. This raises the questions in the motivation of this work.

2) Effectiveness: According to Table 1, in three out of four benchmarks, the ensemble improvements brought by the proposed method are less then 1%. The only exception is only around 15% as well. Hence, the proposed method does not significantly improve the attack effectiveness.

3) Robustness: There is no experiments on evaluating the robustness against purification methods, even the most simple ones like adding Gaussian noise. Based on my understanding, the more similar the adversarial example is to the original image, the more vulnerable it is. Hence, I think the robustness could be a core defect of the proposed method.

4) Typos: there are several typos in the paper, involving noting the second best score *0.510* as the best score on LLAVA in Table 1 and *Stabel Diffusion* in line 259.

**Questions:**

1) What is the motivation of generating adversarial examples highly similar to their original images (rather than just generating some high-fidelity ones)?

2) Could you evaluate the robustness of your attack against purification methods?

---

### Official Review · Reviewer_ioUk · 2024-10-29

**Soundness:** 2
**Presentation:** 3
**Contribution:** 2
**Rating:** 5
**Confidence:** 3

**Summary:**

This paper introduces Adversarial Guided Diffusion (AGD), a method for generating high-fidelity adversarial examples for large language models (LLMs) in multimodal tasks. The method leverages the diffusion model that traditionally generates high-quality images to introduce adversarial noise during back-diffusion sampling. Unlike traditional adversarial attack techniques that usually degrade visual quality, AGD maintains high fidelity by embedding adversarial noise in the diffusion step, preserving the realism of the image while achieving effective model manipulation.

**Strengths:**

1. The paper adopts a strategy of gradually introducing adversarial noise during the inverse sampling process. This method not only enhances the quality of adversarial image generation, but also makes the adversarial noise embed more smoothly into the original image by gradually adjusting the guiding direction of the noise in the diffusion model, thereby reducing the visual damage of the image. This strategy improves the fidelity of the image while ensuring the effectiveness of the attack.
2. The paper introduces edit-friendly inversion to ensure that the generated adversarial image is as close to the target as possible while maintaining the structure of the original image. This strategy is particularly suitable for multimodal tasks with high fidelity requirements, such as visual question answering and image description generation.
3. The paper makes a strategic choice on the time step for introducing adversarial noise, using a truncated diffusion method to reduce the reliance on noise in the early stage of image structure generation. This design effectively balances the attack effect and image fidelity, and shows good robustness in comparative experiments.
4. The paper designs an effective attack strategy in the black-box model environment. By constructing a replacement model with an isomorphic visual encoder, the gradient is used to guide the noise in the reverse sampling stage. This method avoids dependence on the internal parameters of the target model and is suitable for multimodal large models (such as CLIP, etc.) attack scenarios in black-box environments.
5. The paper demonstrates its superiority in indicators such as CLIP score, LPIPS, and SSIM through quantitative comparison with multiple existing methods (such as MF-it, AdvDiffuser, etc.). In addition, qualitative experiments also demonstrate the advantages of the AGD method in image visual quality through multiple sets of adversarial images.

**Weaknesses:**

1. Although the Adversarial-Guided Diffusion (AGD) method proposed in the paper introduces adversarial noise in reverse sampling, it only makes fine-tuning in sampling strategy and editor-friendliness compared with existing diffusion attack methods (such as AdvDiffuser, ACA), and its innovation is limited.
2. The paper selects a specific time step to introduce adversarial noise in the reverse sampling process to improve image fidelity, but the selection and effect of the sampling strategy are only supported by experimental data. I am worried that the lack of theoretical support may affect the universality of the method.
3. The paper uses editor-friendly reverse sampling to improve image quality, but the entire implementation process is complicated, including multiple noise mapping and gradient updates, resulting in high computational costs. In practical applications with limited resources, it is difficult to quickly achieve high efficiency and low latency in adversarial image generation.
4. The paper uses several different models (such as BLIP-2, LLaVA, etc.) for testing, but the selection of these models is limited and does not cover newer, complex multimodal large models. This may lead to doubts about the robustness of the paper's results and the universality of the method.

**Questions:**

Please see weaknesses above.

---

### Official Review · Reviewer_UQGN · 2024-11-02

**Soundness:** 2
**Presentation:** 3
**Contribution:** 2
**Rating:** 5
**Confidence:** 4

**Summary:**

This paper proposes Adversarial-Guided Diffusion, a framework for generating adversarial attacks against multimodal large language models. AGD introduces adversarial noise during the reverse sampling process of conditional diffusion models, using editing-friendly inversion sampling and intelligent timestep selection to create high-fidelity adversarial images that can mislead MLLMs while maintaining image quality.

**Strengths:**

Well-structured experimental validation with clear comparisons to existing methods, supported by visual examples.

**Weaknesses:**

1. This paper does not adequately discuss the computational costs and efficiency of their method with other existing methods. While they mention choosing specific timesteps for optimization, there's no detailed analysis of runtime comparisons, resource requirements, or scalability issues, considering that diffusion models are already computationally intensive.

2. This paper doesn't sufficiently explore how well their method generalizes across different types of MLLMs. The experiments should have included a wider range of models to demonstrate robust generalization, such as the InternLM series, LLaVA-v1.5 series, Qwen-VL series.

3.  This paper fails to provide an evaluation of commercial models, such as  (Gemini, Claude, and GPT-4V, GPT-4o). The paper should also claim specific success rates, and the number of attack attempts for these systems.

**Questions:**

The paper suffers from computational efficiency analysis and generalization capabilities across different models and scenarios.

---

### Note · Authors · 2024-11-12

**Comment:**

We all authors agree to withdraw this paper.

**Withdrawal Confirmation:**

I have read and agree with the venue's withdrawal policy on behalf of myself and my co-authors.